# Presence of small resistant peptides from new *in vitro* digestion assays detected by liquid chromatography tandem mass spectrometry: An implication of allergenicity prediction of novel proteins?

**Rong Wang** *, **Yanfei Wang, Thomas C. Edrington, Zhenjiu Liu, Thomas C. Lee, Andre Silvanovich, Hong S. Moon, Zi L. Liu, Bin Li**

Bayer CropScience, Chesterfield, Missouri, United States of America

* rong.wang@bayer.com

## Abstract

The susceptibility of newly expressed proteins to digestion by gastrointestinal proteases (e.g., pepsin) has long been regarded as one of the important endpoints in the weight-of-evidence (WOE) approach to assess the allergenic risk of genetically modified (GM) crops. The European Food Safety Authority (EFSA) has suggested that current digestion study protocols used for this assessment should be modified to more accurately reflect the diverse physiological conditions encountered in human populations and that the post-digestion analysis should include analytical methods to detect small peptide digestion products. The susceptibility of two allergens (beta-lactoglobin (β-Lg) and alpha-lactalbumin (α-La)) and two non-allergens (hemoglobin (Hb) and phosphofructokinase (PFK)) to proteolytic degradation was investigated under two pepsin digestion conditions (optimal pepsin digestion condition: pH 1.2, 10 U pepsin/μg test protein; sub-optimal pepsin digestion condition: pH 5.0, 1 U pepsin/10 mg test protein), followed by 34.5 U trypsin/mg test protein and 0.4 U chymotrypsin/ mg test protein digestion in the absence or presence of bile salts. All samples were analyzed by sodium dodecyl sulfate–polyacrylamide gel electrophoresis (SDS-PAGE) in conjunction with Coomassie Blue staining and, in parallel, liquid chromatography tandem mass spectrometry (LC-MS) detection. The results provide following insights: 1) LC-MS methodology does provide the detection of small peptides; 2) Peptides are detected in both allergens and non-allergens from all digestion conditions; 3) No clear differences among the peptides detected from allergen and non-allergens; 4) The differences observed in SDS-PAGE between the optimal and sub-optimal pepsin digestion conditions are expected and align with kinetics and properties of the specific enzymes; 5) The new methodology with new digestion conditions and LC-MS detection does not provide any differentiating information for prediction whether a protein is an allergen. The classic pepsin resistance assay remains the most useful assessment of the potential exposure of an intact newly expressed protein as part of product safety assessment within a WOE approach.

**Data Availability Statement:** All relevant data are within the paper and its Supporting Information files.

**Funding:** The authors are employees of Bayer CropScience, a leading manufacturer of crop seeds developed through conventional breeding or biotechnology. The funder had no role in study design, data collection and analysis, decision to publish, or preparation of the manuscript. The specific roles of these authors are articulated in the 'author contributions' section.

**Competing interests:** I have read the journal's policy and the authors of this manuscript have the following competing interest: The authors are employees of Bayer Crop Science, a leading manufacturer of crop seeds developed through conventional breeding or biotechnology. The funder had no role in study design, data collection and analysis, decision to publish, or preparation of the manuscript. This does not alter our adherence to PLOS ONE policies on sharing data and materials.

## Introduction

Assessment of the potential of a newly express protein (NEP) to be allergenic is one component of food and feed safety assessment for genetically modified (GM) foods. There is no single test or parameter that can predict the likelihood a protein or peptide to invoke an allergenic response [1]. Therefore, a weight-of-evidence (WOE) approach is utilized to assess the allergenic potential of a NEP [1–3]. This approach includes an evaluation of the history of safe use of the protein, or closely related homologs, comparisons of the NEP amino acid sequence to those of known allergens, an understanding of the mode-of-action of the NEP, and assessments of the NEP's susceptibility to degradation by gastrointestinal proteases.

The susceptibility of a NEP to gastrointestinal proteases has historically been assessed using an *in vitro* study design incorporated with the naturally occurring digestive protease pepsin and a mixture of pancreatic proteases known as pancreatin [4–6]. Pepsin, the primary digestive enzyme in the stomach, is a broad-spectrum protease that preferentially hydrolyzes peptide bonds containing the aromatic amino acids Phe, Trp, and Tyr at low pH [7, 8]. Pancreatin, whose principal components are trypsin, chymotrypsin, peptidase, lipase, amylase and bile salts, is produced in pancreas and released to the duodenum. Proteases present in the duodenum further break down dietary proteins into small peptides or amino acids to facilitate intestinal absorption. The resistance or susceptibility of a NEP to pepsin degradation was initially linked to allergenicity by a single report [9]; however, this correlation was later found to not be absolute [10–14]. Nevertheless, pepsin resistance tests remain an integral part of the WOE approach to assess allergenic potential of a NEP.

Conditions for a standardized pepsin resistance assay were previously harmonized with proven reproducibility through a multi-laboratory collaboration [5]. The standardized conditions, which specify both the amount of pepsin (10 U of pepsin/mg of test substance) and pH (1.2) for digestion, have been widely accepted by regulatory agencies across the globe for the last 15 years. However, these conditions have also been criticized as not fully representing populations that have impaired or underdeveloped digestive systems where there may be less pepsin or different gastric pH [6]. Additionally, analysis of digestion reactions by staining of sodium dodecyl sulfate–polyacrylamide gel electrophoresis (SDS-PAGE), which was outlined in the standardized pepsin resistance assay, has also been criticized for lacking the ability to detect peptides smaller than 3 kDa. The allergenicity predictability of physiological digestion test conditions that cover the portion of the human population who have impaired digestive conditions (such as people who had gastric bypass surgery or are taking proton inhibitor drugs) remains uncertain because proteins that are readily digested by pepsin using the standardized protocol remain intact or partially intact after incubation of pepsin under sub-optimal conditions [15, 16]. These results indicate that under sub-optimal conditions, high pH and low pepsin-to-protein ratios (PPR), the predictive value of the assay is not improved because both allergens and non-allergens are resistant to pepsin digestion.

The European Food Safety Authority (EFSA) proposes new methodologies for *in vitro* digestion in recently published guidance [17] that includes a proposition to investigate the susceptibility of NEPs to digestion under conditions of high pH and low pepsin:test protein ratios, as is found in atypical human populations. A sequential digestion that includes pepsin for the gastric phase and trypsin/chymotrypsin for the intestinal phase was also suggested. It Is then suggested that the digestion mixtures can then be analyzed by liquid chromatograph-mass spectrometry (LC-MS) to identify peptide fragments $\geq$ 9 amino acids in length that persist throughout the reaction(s), as the EFSA panel hypothesized that more physiological conditions and sensitive peptide detection could provide a clearer distinction between allergens and non-allergens. Presumably, stable peptides from the digestions found in the gastrointestinal tract

could enter the blood stream through intestinal lumen, and may display efficient binding to human leukocyte antigen (HLA) DQ molecules that are typically present in patients with celiac disease (CD) [17]; therefore, these CD peptides might have allergenic potential with undesirable immune responses. Like IgE epitopes that are well documented, CD peptides rich in proline and glutamine are also well characterized. They mainly from four crops (i.e. wheat, barley, rye and oats) and some bacteria [18]. A list of sequences contains CD peptides and their degenerated sequences was proposed by EFSA for sequence analysis [17]. Theoretically, stable fragments could be observed by LC-MS while the relevant peptides that contribute to IgE epitopes and celiac disease can be searched through *in silico* sequence homology analysis without *in vitro* digestion work.

The objective of the present study was to evaluate any improved predictability of EFSA's proposed *in vitro* digestibility methods through the susceptibility of dietary proteins to pepsin and trypsin and chymotrypsin under both optimal and sub-optimal reaction conditions and the presence of small peptides. Two established allergens (beta-lactoglobulin (β-Lg) and alpha-lactalbumin (α-La)) that have been extensively characterized [19–23] and two known non-allergens (hemoglobin (Hb) and phosphofructokinase (PFK)) that have either been commonly used for pepsin activity assays or recommended as a control protein [17] were selected for the present study. A total of four assay conditions were conducted. These included two pepsin conditions (pepsin protein ratio at 10 Unit per mg test protein (PPR 10) and pH 1.2 and pepsin protein ratio at 1 Unit per 10 mg test protein (PPR 0.1) and pH 5.0 for 60-minute incubation) followed by digestion with trypsin and chymotrypsin for 30 min, 60 min, and overnight incubation, in the presence and absence of bile salts. Reaction mixtures were analyzed by SDS-PAGE and LC-MS for the presence of resistant peptide fragments of ≥ 9 amino acids in length (thereafter referred to as peptides ≥ 9 amino acids).

## Materials and methods

### Enzymes. substrate proteins and bile salts

Purified porcine pepsin (Cat. P6887), trypsin (Cat. T7309), chymotrypsin (Cat. CHY5S), hemoglobin (Hb) (Cat. H2625), phosphofructokinase (PFK) (Cat. F0137), alpha-lactalbumin (α-La) (Cat. L5385), and beta-lactoglobulin (β-Lg) (Cat. L3908), sodium taurocholate hydrate (Cat. 86339), sodium glycodeoxycholate (Cat. G9910) were obtained from Sigma Aldrich (St Louis, MO, U.S.A).

### Pepsin degradation followed by trypsin and chymotrypsin degradation

For the digestions (Fig 1), pepsin stock solutions (2000 U/ml in solution, calculated based on certificate of analysis from the vendor, verified in the lab) were prepared in 1% (w/v) NaCl at pH 1.2 or pH 5.0 adjusted with HCl. Substrate protein solutions were prepared daily at a concentration of 2 mg/ml in Milli-Q water. The proteins (Hb, PFK, α-La, and β-Lg) were tested under two separate pepsin assay conditions: optimal pepsin condition (pH 1.2 and PPR 10) and sub-optimal pepsin condition (pH 5.0 and PPR 0.1) [17]. Pepsin and substrate protein mixtures at optimal pepsin condition were incubated in a 37°C water bath for durations of 60 minutes before being quenched and raising pH to ~6.5 by the addition of 0.7 M $Na_2CO_3$ at 10% of the reaction volume and 1 M Tris pH 8 at 5% of the reaction volume. Pepsin and substrate protein mixtures at sub-optimal pepsin conditions were incubated in a 37°C water bath for durations of 60 minutes before being quenched and raising pH to ~6.5 by the addition of 1 M Tris pH 6.5 at 15% of the reaction volume. After the neutralization, samples were then carried out for trypsin and chymotrypsin degradation by addition of trypsin to final 34.5 U/mg test protein, chymotrypsin to final 0.44 U/mg test protein, and $CaCl_2$ to 10 mM final

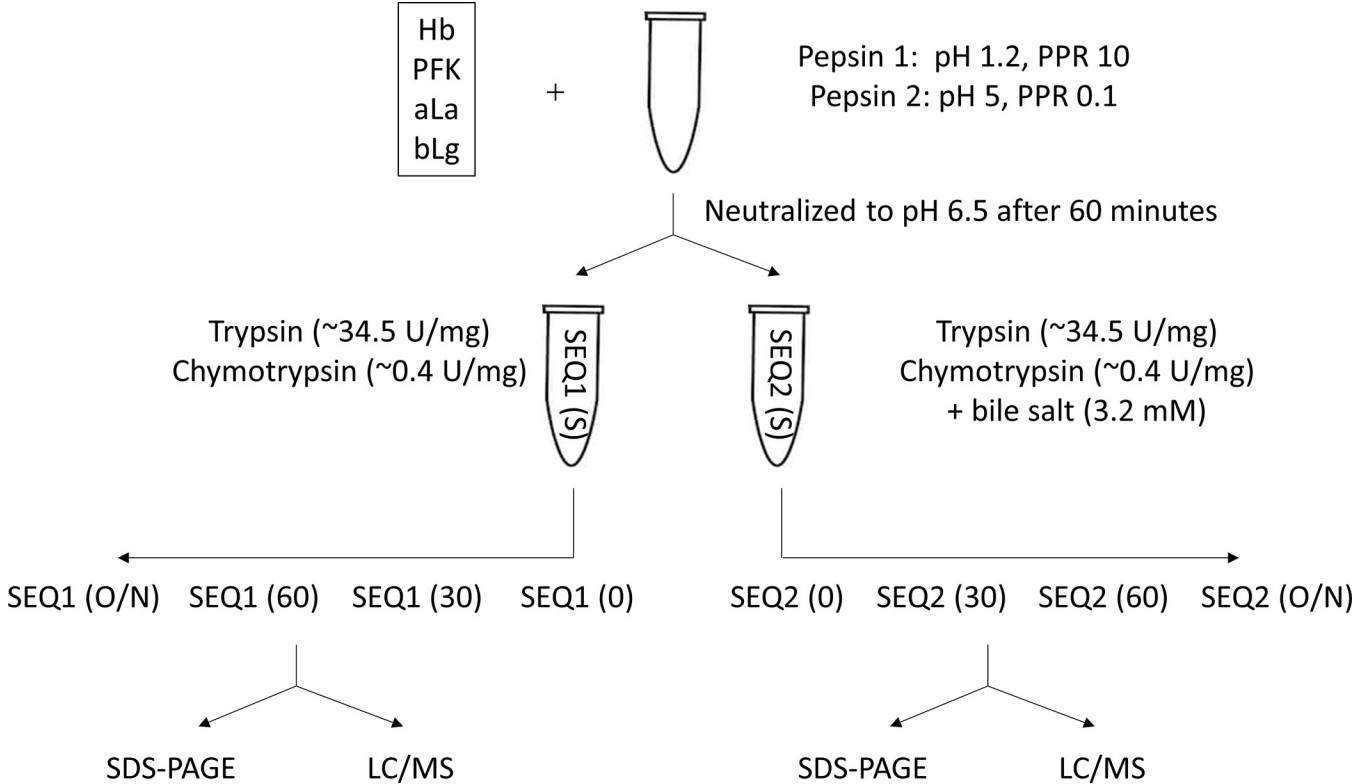

**Fig 1. Sequential digestion work flow.** Each protein was digested by pepsin at pH 1.2 PPR 10 or pH 5 PPR 0.1 followed by trypsin and chymotrypsin in the presence or absence of bile salts for 0 minute, 30 minutes, 60 minutes, or overnight (O/N). Samples were quenched for SDS-PAGE or LC-MS. SEQ stands for sequential digestion.

concentration in a 37˚C water bath for 30 minutes, 60 minutes [17, 19, 20], and overnight (~17 hours). The overnight incubation is a common tryptic mapping condition required for a relatively complete digestion and is used here for comparison. The weight to weight ratios for trypsin and chymotrypsin to test proteins are estimated at about 1:400 and 1:100, respectively. Samples and experimental controls that were absent of enzymes or test proteins were quenched by SDS-PAGE gel loading buffer and heat treatment, followed by -80˚C storage prior to SDS-PAGE analysis. Samples and experimental controls that were absent of enzymes or test proteins were quenched by formic acid at 5% final concentration, followed by -80˚C storage prior to additional sample preparation for LC-MS detection. Triplicate sequential digestions (pepsin digestion followed by trypsin and chymotrypsin digestion) were carried out for all time points except the overnight incubation where duplicates were employed. Duplicate sequential digestions were also carried out with the inclusion of bile salts in the reactions (sodium taurocholate hydrate and sodium glycodeoxycholate to final the concentration of 3.2 mM).

## SDS–PAGE analysis

Samples taken at different time points of the digestion reactions and digestion controls without enzymes or test proteins were subjected to SDS-PAGE utilizing NOVEX[TM] 10–20% (w/v) gradient polyacrylamide Tricine SDS gels separated under constant voltage according to the manufacturer's instructions. NOVEX[TM] Mark 12[TM] molecular weight markers were loaded to estimate/confirm the apparent MWs of proteins and fragments. Following SDS-PAGE, protein

bands or polypeptides were visualized by Coomassie Blue staining and digitally imaged by BIO-RAD GS-900 Densitometer (Hercules, CA, U.S.A.) for visual display.

## LC-MS analysis

Samples treated with trypsin/chymotrypsin post-pepsin digestion were further processed for MS analysis. The samples were desalted using C18 plates according to the manufacturer's protocol (Thermo Fisher Scientific Cat. 60300–426). The desalted samples were dried and subsequently resuspended in 0.1% formic acid prior to MS analysis. Approximately 0.15 μg of each sample (based on the pre-digestion concentration) dissolved in approximately 3 μl was injected into LC-MS.

Peptides were analyzed on an Ultimate 3000 nano LC system using micro flow connected to Orbitrap Fusion MS (Thermo Fisher Scientific, Waltham, MA, USA) operated in Full MS/ ddMS$^2$ IT CID mode. The data were collected with the installed Xcalibur software (Thermo Fisher Scientific, Waltham, MA, USA). The binary mobile phase consisted of buffer A (0.1% formic acid in water) and buffer B (0.1% formic acid in acetonitrile) at a flow rate of 200 μl/ minute with a gradient of 5–30% (v:v) of buffer B for 2 minutes. Chromatography was performed with an Aquasil C18 column (10 × 2.1mm Javelin Guards 6 μm). Full-scan mass spectra were acquired in the Orbitrap over a mass range of 200–1,800 m/z with a resolution of 120,000 at AGC target $2 \times 10^5$. The intense precursor ions between $1.0e^3$ and $1.0e^{20}$ were selected for collision-induced fragmentation with normalized collision energy of 35% with AGC target as $1 \times 10^4$.

Proteins were identified by the Proteome Discoverer (version 1.4; Thermo Fisher Scientific, Waltham, MA, USA). The protein database, including the targeted protein sequences from UniProt and a reversed decoy database, were used for comparison. Data were generated from acquired raw data files with Thermo Xcalibur. The data include only rank 1 peptides and peptides in the top scored proteins. Fragments that derived from enzymes were filtered out. Peptide mass tolerance was set at 10 ppm, fragment mass tolerance was set at 0.6 Da, and peptide charge was set at +2, +3, and +4. False discovery rates for peptide identification of all searches were less than 5.0%. Amino acid sequence of each peptide was confirmed with MS$^2$. Each different peptide was considered as a unique observation. The number of unique peptides that derived from sequentially digested test proteins and equal or greater than 9 amino acids were counted. Triplicate LC/MS analysis were done for digestion conditions without bile salts. Single LC/MS analysis were done for digestion conditions with bile salts. Mean values of unique peptides from those triplicates were reported. Standard deviations from triplicate analysis were generated. The mean values and standard deviations used for error bars in figures are reported in tables as supporting information. The same peptide with and without modification was combined and counted as one observation.

## Results

### Hemoglobin (Hb)

SDS-PAGE followed by Coomassie blue staining analysis demonstrated that intact Hb, a non-allergen, was degraded under pepsin optimal conditions, pH 1.2 PPR 10 for 60 minutes (hereafter, pepsin digestion is mentioned without incubation time since 60-minute is the only incubation time that was used), as evidenced by the complete disappearance of the protein band at ~14 kDa on the SDS-PAGE gel (Fig 2). Additionally, no obvious peptide fragments of Hb were observed in pepsin digestion under optimal conditions. Under sub-optimal conditions, pH 5 PPR 0.1, there was no change in the intensity of the intact Hb band nor were additional bands observed (Fig 2), indicating that pepsin was unable to degrade Hb under these conditions as

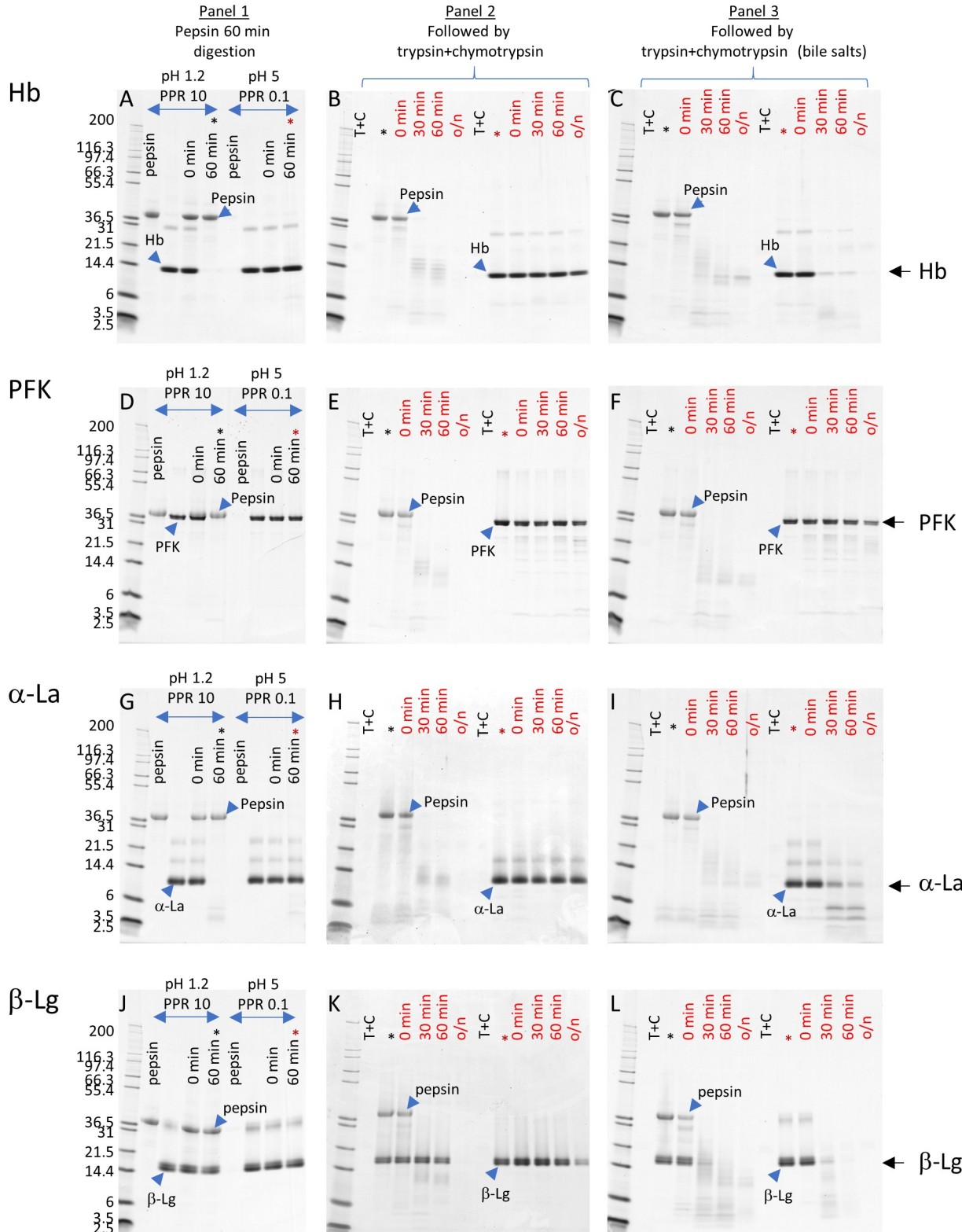

**Fig 2. SDS-PAGE analysis of Hb, PFK, α-La, and β-Lg from *in vitro* digestions.** 1 μg of substrate protein, based upon the pre-degradation concentration, was loaded in each well. **Panel 1 (A, D, G, J)** are samples from gastric digestion. Hb, PFK, and α-La were digested under optimal pepsin condition with low pH while β-Lg had resistant with the presence of intact band at 60 min time point. There is no difference of

susceptibly to sub-optimal pepsin condition among four proteins. **Panel 2 (B, E, H, K)** are samples from subsequent trypsin and chymotrypsin digestion. Intact proteins post pepsin digestions withstood the proteolysis of trypsin and chymotrypsin except β-Lg post sub-optimal pepsin digestion had diminished band intensity after o/n incubation. **Panel 3 (C, F, I, L)** are samples from subsequent trypsin and chymotrypsin digestion with the presence of bile salts. Hb, α-La and β-Lg post sub-optimal pepsin digestion had increased susceptible to trypsin and chymotrypsin with increase incubation time while PFK withstood the proteolysis of trypsin and chymotrypsin.

previously reported [16]. Pepsin was barely visible on the gel due to the amount being 1% of what was in the optimal pepsin condition. In the sequential digestion of Hb samples that were incubated under optimal or sub-optimal conditions, were further processed by a mixture of trypsin and chymotrypsin. As expected, no intact Hb protein band was observed in the samples derived from the optimal pepsin conditions. The intact pepsin protein band (~38 kDa) was degraded at 30 minutes of incubation in trypsin and chymotrypsin resulting in fragments of ~10–14 kDa which disappeared after overnight incubation. Similar results were observed for trypsin and chymotrypsin digestion with the addition of bile salts except for a faint pepsin fragment band at ~10 kDa persisting after overnight incubation.

In the sequential digestion of the Hb samples derived from the sub-optimal pepsin conditions with trypsin and chymotrypsin there was a difference observed due to the presence of bile salts (Fig 2). Without bile salts, there was no difference in the intensity of the intact Hb protein band throughout the time course (30 and 60 minutes, overnight), indicating that trypsin and chymotrypsin were unable to degrade Hb under these test conditions. With bile salts, the intensity of the Hb protein was significantly reduced at the 30- and 60-minute time points and almost completely absent after overnight incubation (Fig 2), indicating that the addition of bile salts improved the ability of trypsin and chymotrypsin to degrade Hb.

LC-MS was used to identify unique Hb peptides ≥ 9 amino acids both after pepsin digestion and sequential digestion with trypsin and chymotrypsin, under both optimal and sub-optimal conditions (Fig 3). Mean values or standard deviations can be found in S1 Table. Triplicate or duplicate analysis were applied for confirmatory observations due to the variability nature of some peptides. After pepsin digestion, an average of 10 and 21 unique Hb peptides ≥ 9 amino acids were observed in the optimal and sub-optimal pepsin conditions, respectively (Table 1). Subsequent digestion with trypsin and chymotrypsin in the absence of bile salts for 60 minutes produced an average of 3 and 17 unique Hb peptides ≥ 9 amino acids in the optimal and sub-optimal pepsin conditions, respectively (Table 1). The number of peptides, 3 vs 17, are clearly linked to the further degradation of intact protein survived in the sub-optimal pepsin condition by the two intestinal enzymes and resulted in more peptides. The persistent and unique sequences of Hb peptides ≥ 9 amino acids produced from overnight sequential digestion without bile salts after incubation of optimal pepsin conditions are listed in Table 3. The addition of bile salts to the trypsin and chymotrypsin digestion reactions reduced the total number of Hb unique peptides ≥ 9 amino acids derived from optimal and sub-optimal pepsin digestions; however, even after overnight sequential digestion, one unique Hb peptide ≥ 9 amino acids was still detected in the optimal pepsin conditions (Table 2).

## Phosphofructokinase (PFK)

SDS-PAGE analysis indicated that intact PFK, a non-allergen, was degraded under pepsin optimal conditions, pH 1.2 PPR 10, as evidenced by the complete disappearance of the protein band at ~35 kDa on the SDS-PAGE gel (Fig 2). Fragments at ~ 3 kDa were observed in the reaction mixture of pepsin digestion, indicating that the peptides were resistant to pepsin digestion under these conditions. Under the sub-optimal conditions, pH 5 PPR 0.1, there was no change in the intensity or protein banding profile of the PFK reaction mixture when

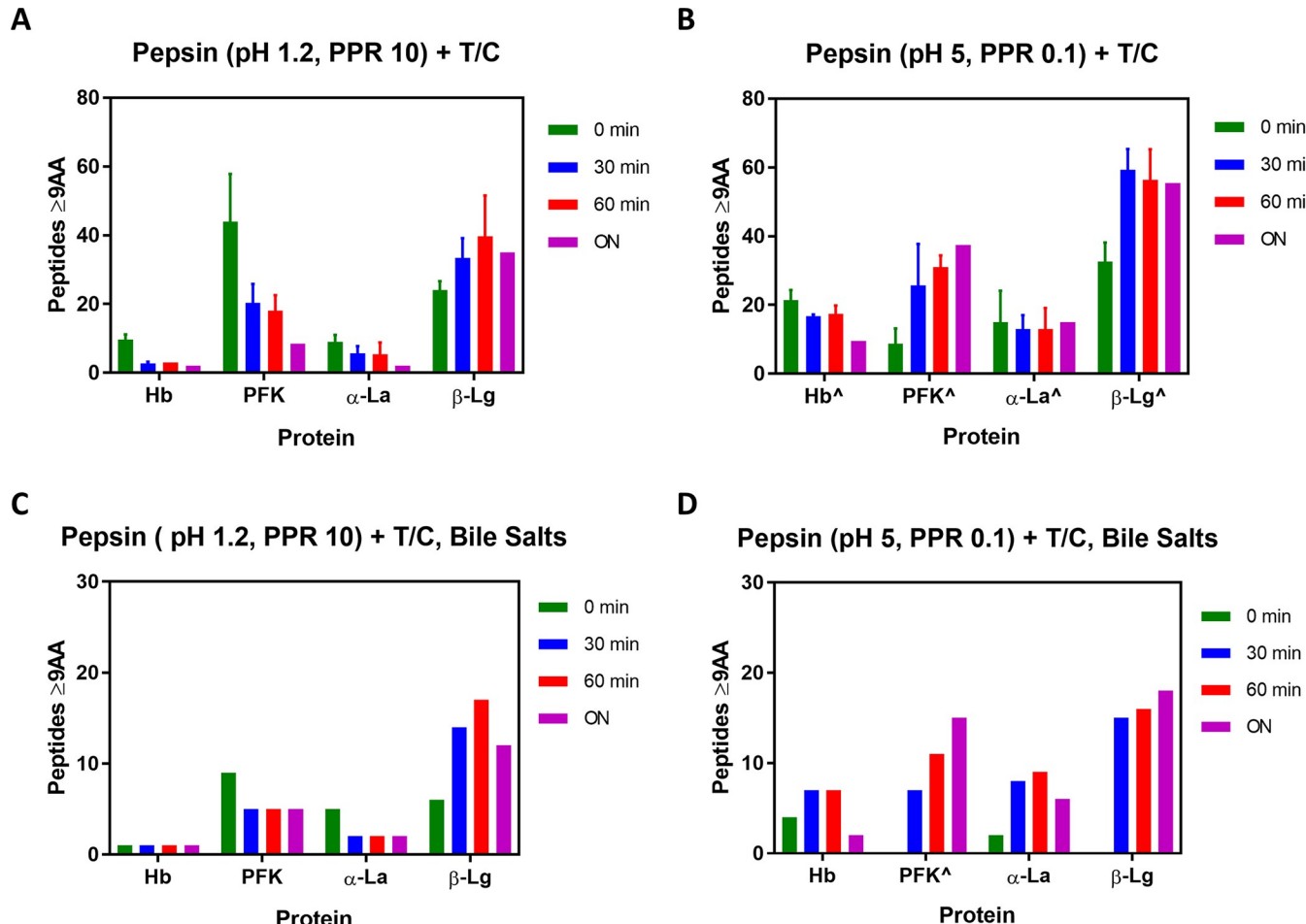

**Fig 3. Number of unique peptides (≥ 9 amino acid) detected and identified by mass spectrometry from sequential digestions for 0, 30, and 60 minutes and overnight.** Green bars represent starting materials for subsequent trypsin and chymotrypsin digestion, which were digested by pepsin for 60 minutes and quenched. Blue bars represent 30 minutes incubation of subsequent trypsin and chymotrypsin digestion. Red bars represent 60 minutes incubation of subsequent trypsin and chymotrypsin digestion. Purple bars represent overnight incubation of subsequent trypsin and chymotrypsin digestion. Error bars represents standard deviations from three independent digestion assays. Data without error bars are from one independent digestion assay. T/C stands for trypsin and chymotrypsin. ^ indicates the presence of an intact protein throughout overnight T/C digestion detected by SDS-PAGE.

analyzed by SDS-PAGE, indicating that pepsin was unable to degrade PFK under these conditions. Pepsin was hardly visible on the gel due to the amount being 1% of the amount in the optimal pepsin condition and close migration distance between PFK and pepsin. There was no observable protein band corresponding to PFK in the SDS-PAGE analysis of the digestion sample from trypsin and chymotrypsin tandem the pepsin optimal condition, as expected (Fig

**Table 1. Number of unique peptides (≥ 9 amino acid) observed under two digestion conditions at four incubation time without bile salts.**

|  | pepsin (pH 1.2, PPR 10) + T/C | | | | pepsin (pH 5, PPR 0.1) + T/C | | | |
|---|---|---|---|---|---|---|---|---|
|  | 0 min | 30 min | 60 min | ON | 0 min | 30 min | 60 min | ON |
| Hb | 10 | 3 | 3 | 2 | 21 | 17 | 17 | 10 |
| PFK | 44 | 20 | 18 | 9 | 9 | 26 | 31 | 38 |
| α-La | 9 | 6 | 5 | 2 | 15 | 13 | 13 | 15 |
| β-Lg | 24 | 33 | 40 | 35 | 33 | 59 | 56 | 56 |

**Table 3. Sequence of unique peptides (≥ 9 amino acid) that were present after overnight digestion of trypsin and chymotrypsin post incubation of optimal pepsin condition without bile salts.**

| Protein Name | Amino Acid Position | MW (kDa) | Peptides Sequences |
|---|---|---|---|
| Hb | 272–281 | 956.55 | QKVVAGVANA |
| | 273–281 | 810.40 | KVVAGVANA |
| PFK | 8–18 | 993.39 | TSGGDSPGMNA |
| | 51–59 | 931.47 | KLEVGDVGD |
| | 98–106 | 866.43 | VVIGGDGSY |
| | 98–108 | 1087.65 | VVIGGDGSYQG |
| | 98–109 | 1122.54 | VVIGGDGSYQGA |
| | 120–129 | 986.48 | VGVPGTIDND |
| | 220–229 | 889.43 | VAEGVGSGVD |
| | 220–230 | 1036.49 | VAEGVGSGVDF |
| | 221–230 | 919.34 | AEGVGSGVDF |
| | 222–230 | 866.39 | EGVGSGVDF |
| α-La | 40–49 | 1090.50 | AIVQNNDSTE |
| | 109–117 | 1006.15 | ALCSEKLDQ |
| β-Lg | 9–20 | 1350.70 | GLDIQKVAGTWY |
| | 20–32 | 1354.70 | YSLAMAASDISLL |
| | 21–32 | 1191.63 | SLAMAASDISLL |
| | 21–32 | 1207.63 | SLAmAASDISLL |
| | 21–39 | 1873.96 | SLAMAASDISLLDAQSAPL |
| | 25–39 | 1471.76 | AASDISLLDAQSAPL |
| | 40–59 | 2341.71 | RVYVEELKPTPEGDLEILLQ |
| | 40–60 | 2469.36 | RVYVEELKPTPEGDLEILLQK |
| | 41–57 | 1944.02 | VYYEELKPTPEGDLEIL |
| | 41–59 | 2185.17 | VYYEELKPTPEGDLEILLQ |
| | 41–60 | 2313.26 | VYYEELKPTPEGDLEILLQK |
| | 41–61 | 2499.34 | VYYEELKPTPEGDLEILLQKW |
| | 42–54 | 1489.74 | YVEELKPTPEGDL |
| | 42–55 | 1618.78 | YVEELKPTPEGDLE |
| | 43–54 | 1326.68 | VEELKPTPEGDL |
| | 43–55 | 1455.72 | VEELKPTPEGDLE |
| | 43–57 | 1681.89 | VEELKPTPEGDLEIL |
| | 43–58 | 1794.97 | VEELKPTPEGDLEILL |
| | 43–59 | 1923.04 | VEELKPTPEGDLEILLQ |
| | 43–60 | 2051.13 | VEELKPTPEGDLEILLQK |
| | 43–61 | 2237.21 | VEELKPTPEGDLEILLQKW |
| | 45–55 | 1227.61 | ELKPTPEGDLE |
| | 46–55 | 1098.57 | LKPTPEGDLE |
| | 46–56 | 969.53 | LKPTPEGDL |
| | 83–91 | 1044.57 | KIDALNENK |
| | 83–93 | 1256.72 | KIDALNENKVL |
| | 84–93 | 1128.63 | IDALNENKVL |
| | 91–116 | 3064.55 | KVLVLDTDYKKYLLFCMENSAEPEQS |
| | 92–100 | 1065.58 | VLVLDTDYK |
| | 93–100 | 1094.61 | LVLDTDYKK |
| | 92–101 | 1193.68 | VLVLDTDYKK |
| | 123–131 | 1059.50 | VRTPEVDDE |
| | 123–132 | 1130.53 | VRTPEVDDEA |
| | 123–135 | 1500.76 | VRTPEVDDEALEK |
| | 123–136 | 1647.82 | VRTPEVDDEALEKF |
| | 123–138 | 1890.95 | VRTPEVDDEALEKFDK |
| | 125–135 | 1245.58 | TPEVDDEALEK |
| | 125–136 | 1392.65 | TPEVDDEALEKF |
| | 125–138 | 1635.78 | TPEVDDEALEKFDK |
| | 128–138 | 1308.63 | VDDEALEKFDK |
| | 129–138 | 1209.56 | DDEALEKFDK |
| | 149–162 | 1658.87 | LSFNPTQLEEQCHI |
| | 150–162 | 1545.71 | SFNPTQLEEQCHI |

2). Similar to the corresponding samples from the Hb digestion reaction, a pattern of protein bands corresponding to pepsin and pepsin peptides was observed for the tandem digestion samples with or without bile salts.

There was no difference observed between the absence and presence of bile salts in the sequential digestion of the PFK samples derived from the sub-optimal pepsin conditions with trypsin and chymotrypsin (Fig 2). The intensity of the intact PFK protein band was not significantly reduced throughout the time course (30 and 60 minutes, overnight), which indicated that trypsin and chymotrypsin were unable to efficiently degrade PFK under these test conditions. In the presence of bile salts, the intensity of the intact PFK protein band was slightly reduced after overnight incubation and the pattern of faint peptide bands was similar to the corresponding digestion samples without bile salts. These results indicate that the addition of bile salts may have slightly improved the ability of trypsin and chymotrypsin to degrade PFK.

LC-MS identified unique PFK peptides $\geq 9$ amino acids both after pepsin digestion and after subsequent digestion with trypsin and chymotrypsin (Fig 3). After pepsin digestion, an average of 44 and 9 unique PFK peptides $\geq 9$ amino acids were observed in the optimal and sub-optimal pepsin conditions, respectively (Table 1). Sub-optimal pepsin digestion resulted in less peptides likely due to low hydrolysis. Subsequent digestion with trypsin and chymotrypsin in the absence of bile salts for 60 minutes produced an average of 18 and 31 observable unique PFK peptides $\geq 9$ amino acids derived from the optimal and sub-optimal pepsin conditions, respectively (Table 1). The large number of average 31 peptides presents the same phenomenon occurred to Hb due to further degradation of intact protein. The unique sequences of the PFK peptides $\geq 9$ amino acids that remained after digestion by pepsin under optimal conditions followed digestion with trypsin and chymotrypsin overnight in the absence of bile salts are listed in Table 3. The addition of bile salts to the trypsin and chymotrypsin digestion reactions reduced the total number of unique PFK peptides $\geq 9$ amino acids observable by LC-MS; however, even after allowing the second digestion reaction to proceed overnight there remained 5 and 15 observably unique PFK peptides $\geq 9$ amino acids in the optimal and sub-optimal pepsin conditions, respectively (Table 2).

## Alpha-lactalbumin (α-La)

SDS-PAGE analysis indicated that intact α-La, a known allergen, was degraded under pepsin optimal conditions, pH 1.2 PPR 10, as evidenced by the complete disappearance of the protein band at ~13 kDa on the SDS-PAGE gel (Fig 2). A smaller peptide band at ~4 kDa was observed in the reaction mixture of pepsin digestion, indicating that these peptides were resistant to pepsin digestion under these conditions. Under the sub optimal conditions, pH 5 PPR 0.1, there was no change in the intensity of the intact α-La protein band after 60 minutes of incubation. These results indicated that pepsin was unable to efficiently degrade intact α-La under these conditions.

As expected, there was no observable protein band corresponding to intact α-La in the SDS-PAGE analysis of the digestion sample from optimal pepsin condition followed by trypsin

**Table 2. Number of unique peptides ($\geq$ 9 amino acid) observed under two digestion conditions at four incubation time with bile salts.**

|  | pepsin (pH 1.2, PPR 10) + T/C/bile salts | | | | pepsin (pH 5, PPR 0.1) + T/C/bile salts | | | |
|---|---|---|---|---|---|---|---|---|
|  | 0 min | 30 min | 60 min | ON | 0 min | 30 min | 60 min | ON |
| Hb | 1 | 1 | 1 | 1 | 4 | 7 | 7 | 2 |
| PFK | 9 | 5 | 5 | 5 | 0 | 7 | 11 | 15 |
| α-La | 5 | 2 | 2 | 2 | 2 | 8 | 9 | 6 |
| β-Lg | 6 | 14 | 17 | 12 | 0 | 15 | 16 | 18 |

and chymotrypsin, (Fig 2). Similar to the corresponding samples from the Hb and PFK digestion reactions, a pattern of protein bands corresponding to pepsin and pepsin peptides was observed for the sequential digestion samples with or without bile salts.

In the sequential digestion of the α-La samples derived from the sub-optimal pepsin conditions with trypsin and chymotrypsin there was a difference observed due to the presence of bile salts (Fig 2). In the absence of bile salts, the intensity of the intact α-La protein band was not significantly reduced throughout the time course (30 and 60 minutes, overnight). These results indicated that trypsin and chymotrypsin was unable to efficiently degrade intact α-La under these test conditions. In the presence of bile salts, the intensity of the intact α-La protein band was significantly reduced after 60 minutes of incubation and a pattern of smaller peptide bands was observable for 60 min and faintly observable after overnight incubation. These results indicate that the addition of bile salts improved the ability of trypsin and chymotrypsin to degrade intact α-La.

LC-MS analysis of the α-La digestion reactions identified peptides ≥ 9 amino acids both after pepsin digestion and after subsequent digestion with trypsin and chymotrypsin (Fig 3). After pepsin digestion, an average of 9 and 15 α-La peptides ≥ 9 amino acids were observed in the optimal and sub-optimal pepsin conditions, respectively (Table 1). Subsequent digestion with trypsin and chymotrypsin in the absence of bile salts for 60 minutes produced an average of 5 and 13 observable unique α-La peptides ≥ 9 amino acids in the optimal and sub-optimal pepsin conditions, respectively. The large number of 13 peptides shows the same phenomenon occurred to Hb and PFK due to further degradation of intact protein. The unique sequences of the α-La peptides ≥ 9 amino acids that remained after digestion by pepsin under optimal conditions followed digestion with trypsin and chymotrypsin overnight in the absence of bile salts are listed in Table 3. The addition of bile salts to the trypsin and chymotrypsin digestion reactions reduced the total number of unique α-La peptides ≥ 9 amino acids observable by LC-MS, however, even after allowing the second digestion reaction to proceed overnight there remained 2 and 6 observable α-La peptides ≥ 9 amino acids in the optimal and sub-optimal pepsin conditions, respectively (Table 2).

## Beta-lactoglobulin (β-Lg)

SDS-PAGE analysis indicated that intact β-Lg, a known allergen, was resistant to pepsin degradation under the pepsin optimal conditions, pH 1.2 PPR 10, as evidenced by the persistence of a double band at ~16 kDa throughout the time course on the SDS-PAGE gel (Fig 2). Likewise, under the sub optimal pepsin conditions, pH 5 PPR 0.1, there is no change in the intensity of the major β-Lg protein bands. These results indicate that pepsin was unable to efficiently degrade intact β-Lg under either the optimal or sub optimal conditions. The reaction mixtures under both conditions were then incubated with chymotrypsin and trypsin for 30 minutes, 60 minutes, and overnight in the presence or absence of bile salts.

The major β-Lg protein bands corresponding to intact β-Lg that were observed under both optimal and sub optimal pepsin digestion conditions remained persistent throughout the time course of subsequent incubation with trypsin and chymotrypsin in the absence of bile salts (Fig 2). These results indicate that in the absence of bile salts, trypsin and chymotrypsin are unable to completely degrade intact β-Lg after overnight incubation. Like the corresponding samples from the Hb, PFK and α-La digestion reactions, a pattern of protein bands corresponding to pepsin and pepsin peptides was observed for the tandem digestion samples with or without bile salts.

In the sequential digestion of the β-Lg samples derived from the sub-optimal pepsin conditions with trypsin and chymotrypsin, the intact β-Lg was significantly reduced in the presence

of bile salts (Fig 2). These results indicate that the addition of bile salts significantly improved the degradation of intact β-Lg by trypsin and chymotrypsin.

LC-MS analysis identified unique β-Lg peptides ≥ 9 amino acids both after pepsin digestion and after subsequent digestion with trypsin and chymotrypsin (Fig 3). After pepsin digestion, an average of 24 and 32 unique β-Lg peptides ≥ 9 amino acids were observed in the optimal and sub-optimal pepsin conditions, respectively (Table 1). Subsequent digestion with trypsin and chymotrypsin in the absence of bile salts for 60 minutes produced an average of 40 and 56 observable unique β-Lg peptides ≥ 9 amino acids in the optimal and sub-optimal pepsin conditions, respectively. The large number of average 56 peptides displays the same phenomenon occurred to previous three proteins due to further degradation of intact protein. The unique sequences of the β-Lg peptides that remained after digestion by pepsin under optimal conditions followed digestion with trypsin and chymotrypsin overnight in the absence of bile salts are listed in Table 3. The addition of bile salts to the trypsin and chymotrypsin digestion reactions reduced the total number of unique β-Lg peptides ≥ 9 amino acids observable by LC-MS; however, even after allowing the subsequent digestion to proceed overnight there remained 12 and 18 observable unique β-Lg peptides ≥ 9 amino acids in the optimal and sub-optimal pepsin conditions, respectively (Table 2).

## Discussion

The determination of the susceptibility of proteins to pepsin and/or pancreatin has been historically utilized to mimic the digestion of the proteins under conditions that the proteins will encounter in the human digestive tract. An *in vitro* digestibility assessment can provide some insight into the likelihood of systemic exposure to structurally intact and biologically active proteins, thus informing the safety of a protein for human and animal consumption. However, the physiological conditions of *in vivo* protein digestion are highly complex and constantly changing, such that no single *in vitro* model can fully represent, or simulate, the *in vivo* digestion processes [6]. For NEPs introduced into food crops, the susceptibility of the proteins has been assessed using these conditions with purified protein since 2003 as part of the WOE that assesses the safety of an introduced NEP in terms of exposure and the possibility that it can persist and be absorbed and cause an allergic reaction [1]. It is impossible to assess a real digestion scenario in human, in which huge detection challenges exist for NEPs. First, the proteins are usually expressed at very low level in genetic modified crops (e.g. mg of protein per kg of grain). Second, the proteins are usually lost their structure and function due to food processing (e.g. heat or alkaline limewater). Third, very sensitive detection is needed to find the proteins in food matrix. Recently, EFSA published new guidance for assessing the allergenic potential of NEPs where the proposed *in vitro* digestion assay conditions that included varying the pH and pepsin-to-protein ratios, followed by sequential digestion with two intestinal enzymes, trypsin and chymotrypsin were proposed [17]. This guidance also suggested the use of LC-MS to identify peptides ≥ 9 amino acids that remain after the time course of the digestion reactions. EFSA noted that such peptides may be presented to human immune systems via the GI-tract and potentially yield adverse IgE and non-IgE-mediated reactions. The key question is whether peptide detection is relevant to allergenicity prediction of a new protein.

Peptides are expected with the three enzymes used in this proposed assessment. Pepsin, trypsin and chymotrypsin, have well-characterized biochemical and kinetic properties; optimal reaction pH and kinetic properties [24]. All enzymes have an optimal pH, where their kinetic activity is at its most efficient. Changing that pH in reaction media, either more acidic or basic, will impact the ability of the enzyme to catalyze its specific reaction [25]. Pepsin has an optimum pH of 1–2 [8]; therefore, it was inevitable to have less effective digestion in all of the

proteins tested at higher pH. Even at its optimal pH, pepsin doesn't completely hydrolyze a protein due to lower efficiency of hydrolysis of peptides than that of an intact protein [26]. Each enzyme utilized has a corresponding specificity or preference for hydrolyzing specific peptide bonds based upon amino acid sequence. Pepsin will cleave peptide bonds preferably adjacent to aromatic or hydrophobic amino acids [27]. Trypsin and chymotrypsin will cleave peptide bonds at carboxyl side of lysine (K), arginine (R), phenylalanine (F), tryptophan (W), and tyrosine (Y) amino acids [28]. Neighboring amino acid such as proline and acidic amino acids are known to impact the cleavage site and result in missed cleavage [29, 30]. Based upon these established specificities and the variety of dietary protein sequences, it is likely that pepsin, trypsin and chymotrypsin will not cleave all peptide bonds, and, as a result these *in vitro* assays would yield peptide products ≥ 9 amino acids. Even so, the sequential digestion clearly demonstrated that there is no absolute correlation between intact protein exposure and allergenicity since PFK, a protein with an absolute history of safe use and a known non-allergen, was resistant to degradation under sub-optimal sequential digestion. Different or overlapping IgE epitopes of α-La and β-Lg were reported [31–35]. The observed peptides of α-La and β-Lg from more extensive digestion, the optimal pepsin followed by trypsin and chymotrypsin digestion, do not match with the IgE epitopes. It is important to come back at macro level with SDS-PAGE analysis where the presence of intact protein serves a good indication of available IgE epitopes (Fig 4). The detected peptides do not match known IgE epitopes. Additionally, there were no matches to CD peptides or their degenerated sequences when the sequences of all four test proteins were compared to the EFSA recommended CD peptide list. The question is what the relevance of peptide detection is and how the observed peptides serve a purpose when an intact protein or large fragments can be detected by SDS-PAGE. From a risk-assessment perspective, *in vitro* digestibility of a protein is currently relevant for exposure assessment before more research leads to a clear understanding of allergenic sensitization and elicitation.

Computer based analysis of the potential pepsin cleavage sites in the four test proteins (Hb, PFK, α-La and β-Lg) confirmed that there will likely be peptide products ≥9 amino acids even under optimal digestion conditions (Table 4). When these resulting peptide products are further digested by trypsin and chymotrypsin, which as noted above have relatively higher specificity than pepsin, further degradation will not occur if those specific amino acids are not present in the peptide sequence. This hypothesis was confirmed since peptides products ≥ 9 amino acids that lack specific trypsin and chymotrypsin sites persisted throughout the course of sequential digestion (Table 3). Specifically, peptide products with the amino acid sequences of QKVVAGVANA and KVVAGVANA from Hb and AIVQNNDSTE and ALCSEKLDQ from α-La, which were predicted *in silico* by PeptideCutter, are unique peptides present after sequential digestion with pepsin followed by a mixture of trypsin and chymotrypsin. The miscleavage of trypsin was related to the acidic residues on either side of the cleavage site, which significantly slow down the hydrolysis (30) as well as to sub-optimal pH at 6.5 for trypsin and chymotrypsin [36]. Therefore, even without any experimental investigation, it is reasonable to predict that digestion of a protein by pepsin followed by trypsin and chymotrypsin under sub-optimal conditions would yield peptide fragments ≥ 9 amino acids regardless of the identity of the protein. This also points to the fact that assessing the susceptibility of a NEP to pepsin, trypsin and chymotrypsin is of limited utility in predicting the allergenicity based on "digestibility". Varying the conditions do not clearly contribute to the overall WOE of the safety of a NEP.

Peptides are expected also due to the kinetics of digestive enzymes. It is well known that the rate that an enzyme catalyzes a reaction and yields a product is impacted by the concentration of the substrate and enzyme. Under conditions of saturating substrate, the apparent rate of the

reaction will be directly proportional to the enzyme to substrate ratio. The greater the amount of enzyme, the more product that is produced in a given period of time. The ratio of trypsin to the protein substrates in the assay conditions tested herein is about 1:400 (w:w, trypsin:protein substrate), which is much less than the 1:20 to 1:100 (w:w) ratios used in tryptic digestions for proteomics analyses [37]. For proteomics analysis, tryptic digestions are typically incubated overnight at pH 8.0 to ensure complete digestion. Based upon trypsin and chymotrypsin kinetics, it is reasonable to predict that the conditions tested herein, incubation with trypsin and chymotrypsin for 60 minutes at a significantly lower enzyme:protein ratio, would result in less target protein or peptide digestion when compared to typical conditions used for proteomic analysis. The experimental results described in Table 1 indicate that a large number of unique peptides ≥9 amino acids detected after sequential digestion, which could at least partially be attributed to overabundance of substrate relative to trypsin and chymotrypsin as reflected in the trypsin:protein ratio [38]. This incomplete sequential digestion (i.e., pepsin followed by trypsin and chymotrypsin) is even further exacerbated when sub-optimal pepsin conditions (e.g., pH 5) are used. Under such an experimental scheme the presence of unique stable peptides (≥ 9 amino acids) from subsequent trypsin and chymotrypsin digestion was noticeably higher, as expected (Table 1).

Dietary proteins have intrinsic characteristics that make them more stable under certain digestion conditions which in turn may also affect their susceptibility to pepsin, trypsin and chymotrypsin degradation. For example, α-La contains 4 pairs of disulfide bonds which should make it more stable, yet it unfolds at pH 2, permitting greater accessibility for digestive proteases [39]. This is in stark contrast to β-Lg, which contains only one pair of disulfide bonds yet retains its native conformation at pH 1 [40]. These intrinsic properties may explain why only two unique stable peptides ≥9 amino acids were observed for α-La compared to 35 stable peptides ≥9 amino acids observed for β-Lg after overnight sequential digestion post optimal pepsin digestion (Table 1). The influence of intrinsic protein stability was also observed between the two non-allergenic proteins, Hb and PFK. Hb is more susceptible to acid denaturation than PFK [41, 42]. As a result, PFK had 5 times more unique stable peptides ≥9 amino acids when compared to Hb after 60-minute incubation of trypsin and chymotrypsin post optimal pepsin digestion. Bile salts are major organic components in bile acids while phosphatidylcholine present at low concentration on human gastric compartment [43]. Bile salts facilitated degradation and reduced the number of unique peptides across all four proteins (Table 2, Fig 3). They appear to have a destabilizing effect on the proteins tested due to their non-denaturing surfactant properties [44]. While they don't affect trypsin and chymotrypsin activity, they do inhibit pepsin activity [45]. Therefore, Bile salts were only tested along with trypsin and chymotrypsin in present study considering bile salts are release from bile duct into upper intestine [46]. Bile salts also can be ionized and lead to signal suppression during MS peptide detection [47]. Trace amount of bile salts could still be present after repeated desalting steps and interfere ionization. Regardless of the surfactant influence on the intrinsic stability of a test protein or ion suppression effect due to the presence of bile salts, unique stable peptides ≥ 9 amino acids are present across all four proteins from four digestion conditions.

State-of-the-art, high resolution LC-MS has become a familiar technology for the characterization of food proteins and peptides [48]. With such a high degree of sensitivity, many dietary peptides have been observed in human's blood stream and milk [49, 50]. Digestion resistant peptides from gluten have a relatively clear threshold for sensitization or elicitation of celiac disease of 20 ppm according to FDA food labelling [51]. The high resolution LC-MS can easily identify trace amount of peptides at the level below this threshold, which is considered as a no-observed-adverse-effect level [52]. In the current study, detection sensitivity of the LC-MS method was assessed. A similar number of stable peptides were identified when fraction of

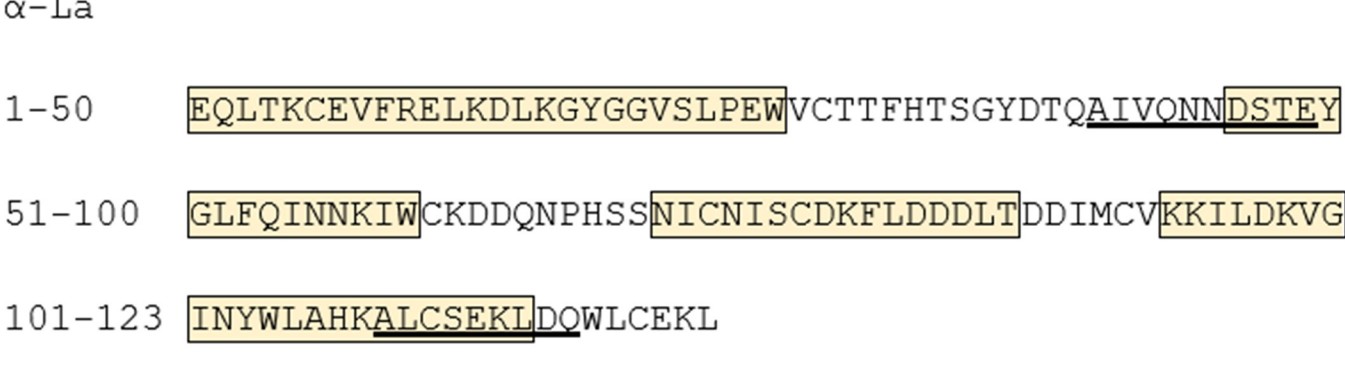

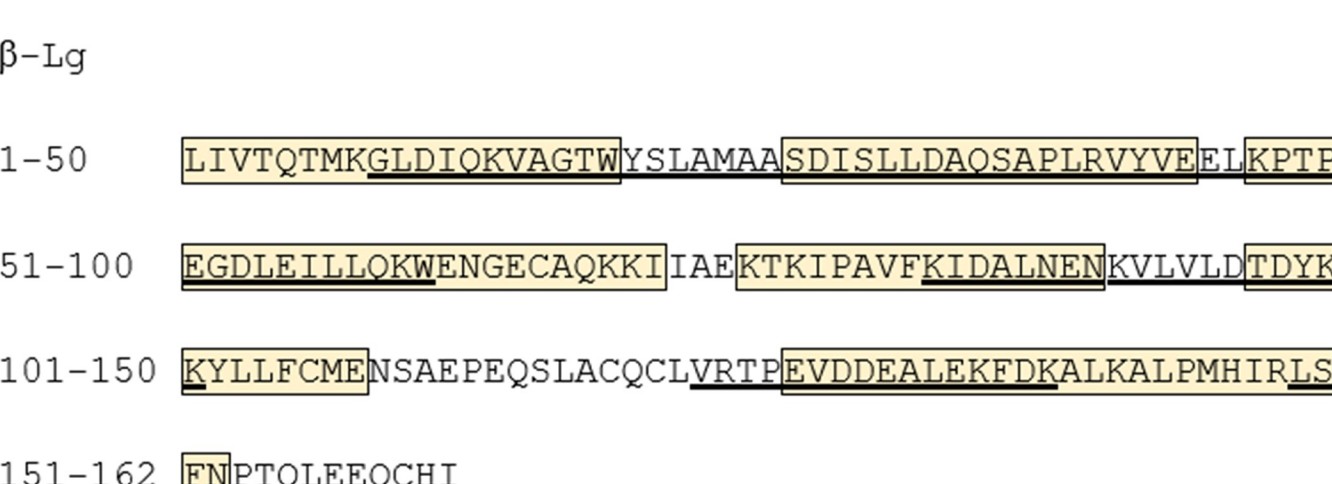

**Fig 4. Comparison of IgE epitopes and observe peptides (length ≥ 9 amino acid) of α-La and β-Lg.** Underlined sequences are observed stable peptides (≥ 9 amino acid) of α-La and β-Lg from overnight digestion of trypsin and chymotrypsin post incubation of optimal pepsin condition without bile salts. Boxed amino acids are reported IgE epitopes.

samples, such as 1/100 of a sequential digestion, were injected into LC-MS with more sensitive ionization method as nano LC-MS (Table 2, supporting information), which corresponds to detection at ~0.5 ppm concentration (calculated as an intact starting material). Similar observation was reported from a non-peer reviewed publication [53], in which many stable small peptides can be detected from *in vitro* digestions. When such a sensitive method is not capable to quantify various unique peptide, the detection of peptides is achievable discounting peptide ionization efficiency. However, the presence of peptides from every digestion condition regardless of presence of intact proteins confound the ability to provide a clear distinction between digestible and non-digestible proteins. One proven method is SDS-PAGE, which as demonstrated can clearly differentiate β-Lg from Hb, PFK, and α-La using conditions based upon the classic pepsin degradation assay.

The present study evaluated the new digestion methodology proposed by EFSA for the allergenicity assessment of genetically modified (GM) plants. Of the four proteins used in this evaluation, three proteins, Hb, PFK, and α-La, showed to be labile under optimal pepsin digestion conditions, but were resistant to sub-optimal digestion conditions. Stable peptides ≥ 9 amino acids were observed in all proteins under all conditions tested. Especially, digestion of the known

**Table 4. Four test proteins represent diverse characteristics of physiochemical properties that could affect pepsin hydrolysis.**

| Protein Name | Molecular Weight (kDa)[1] | Theoretical pI[1] | Number of Disulfide Bonds | Number of Aromatic Amino Acids in Total (Percent of Total)[1] | Estimated Number of peptides (length ≥ 9 amino acid)[2] |
|---|---|---|---|---|---|
| Hemoglobin (Hb) Subunit α | 15.04 | 9.26 | 0 | 10/141 (7%) | 3 |
| Hemoglobin (Hb) Subunit β | 16.03 | 7.98 | 0 | 12/146 (8%) | 3 |
| Phosphofructokinase (PFK) | 34.1 | 6.37 | 1 | 17/319 (5%) | 5 |
| alpha lactalbumin (α-La) | 14.2 | 5.11 | 4 | 12/123 (10%) | 12 |
| beta lactoglobulin (β-Lg) | 18.2 | 5.07 | 1 | 10/162 (6%) | 5 |

[1] Calculated based on amino acid sequences using CLC Genomic Workbench (QIAGEN).

[2] Estimated pepsin cleavage sites at pH 1.3 using PeptideCutter (http://web.expasy.org/peptide_cutter/).

allergen β-Lg under the conditions tested produced similar results to those observed for the non-allergen PFK when evaluated by LC-MS. Results from this study indicate that while the known allergen β-Lg is resistant to sequential digestion by optimal or sub-optimal pepsin digestion followed by trypsin and chymotrypsin, the presence of bile salts destabilized the protein and made it susceptible to sequential digestion when evaluated by SDS-PAGE. The utilization of LC-MS intended to support assessment of the resistance of dietary proteins to gastrointestinal digestion could not provide data to support a clear distinction between allergenic and digestible proteins when evaluated based upon the presence of unique apparently stable peptides ≥9 amino acids.

The observations reported are aligned with a recent debate towards the pending new guidance document on *in vitro* digestibility provided by a joint initiative of ImpARAS and INFO-GEST regarding its relevance towards the allergenicity assessment of novel proteins [14]. The overly simplified physiological digestion conditions through so-called refinement can neither truly reflect physiological conditions nor provide more reliable knowledge about the susceptibility of a NEP in the gastrointestinal tract [54]. The classic pepsin digestion condition can distinguish digestible proteins from non-digestible proteins. Therefore, despite the lack of representation of dynamic physiological conditions, the classical pepsin assay and SDS-PAGE analysis are applicable and provides useful information on the likelihood of exposure to a structurally intact and biologically active proteins as part of WOE approach.

## Supporting information

**S1 Data.**
(XLSX)

**S1 Table.**
(XLSX)

**S2 Table.**
(XLSX)

**S1 Raw Image.**
(TIF)

## Acknowledgments

The authors would like to thank Dr. Luis Burzio for his helpful comments to this manuscript. The authors would like to thank Jill Horn for the assistance of LC-MS data process.

## Author Contributions

**Conceptualization:** Rong Wang.

**Data curation:** Rong Wang, Yanfei Wang.

**Methodology:** Rong Wang, Yanfei Wang.

**Writing – original draft:** Rong Wang, Yanfei Wang, Thomas C. Edrington.

**Writing – review & editing:** Rong Wang, Yanfei Wang, Thomas C. Edrington, Zhenjiu Liu, Thomas C. Lee, Andre Silvanovich, Hong S. Moon, Zi L. Liu, Bin Li.

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
