## [Decision Letter · Decision Letter 0]

1 Apr 2020

PONE-D-20-05235

Presence of small resistant peptides from new in vitro digestion assays detected by liquid chromatography-tandem mass spectrometry: an implication of allergenicity prediction of novel proteins?

PLOS ONE

Dear Dr Ron Wang,

Thank you for submitting your manuscript to PLOS ONE and the time taking its review. The reviewer, who I agree with, has had problems in loading the comments and that is the reason I am copying them. We feel that it has merit but does not fully meet PLOS ONE’s publication criteria as it currently stands. Therefore, we invite you to submit a revised version of the manuscript that addresses the points raised during the review process.

"The study provided by the authors is according to state of the art and provides “real life data” comparing 2 allergens versus 2 non-allergenic proteins using the digestion assays with pepsin and chymotrypsin with and without bile salts following the EFSA Guidance document 2017. In their comparative analysis the authors compare 2 allergens and 2 non-allergenic proteins and follow their degradation via different conditions. This guidance document from EFSA J. 2017 states the refined /extended digestion assays are implemented for a time of 2 years to collect data that either support this part of weight of evidence approach or not – so this is a valuable information in that context.

The study is valid and highly relevant, state of the art, and provides data for the risk assessment discussion while it does not provide novel aspects.

The authors conclude that using different protocols for digestion assays and subsequent LC-MS detection does not provide additional information regarding allergenic risk assessment. Their conclusions is to stay with the established pepsin digestion assay which is sufficient for a weight of evidence approach for safety assessment of potential allergens.

What is missing is a discussion on peptides relevant for celiac disease where a) a library of peptides is available and b) the restriction of HLA types allows a thorough search for respective peptides. Has this been performed – or can this be discussed accordingly? Again this would be of relevance for the EFSA interim phase -see guidance document 2017. The discussion focuses on the enzyme kinetics and optimal conditions – true and relevant however, for physiological conditions  in humans it may be a different scenario – this should be reflected in the discussion. Also the aspect of a potential protective role of phosphatidylcholine for digestion (Ref. Mandalari et al) should be discussed. Furthermore, as presented in the guidance document aspects on celiac disease and the relevant peptides should be included in the discussion.

Minor criticism:

What was the underlying rationale of selecting the non-allergenic proteins"

We would appreciate receiving your revised manuscript by May 15th. To enhance the reproducibility of your results, we recommend that if applicable you deposit your laboratory protocols in protocols.io, where a protocol can be assigned its own identifier (DOI) such that it can be cited independently in the future. For instructions see: http://journals.plos.org/plosone/s/submission-guidelines#loc-laboratory-protocols

We look forward to receiving your revised manuscript.

Kind regards,

Maria Gasset, Ph.D.

Academic Editor

PLOS ONE

Additional Editor Comments (if provided):

The study provided by the authors is according to state of the art and provides “real life data” comparing 2 allergens versus 2 non-allergenic proteins using the digestion assays with pepsin and chymotrypsin with and without bile salts following the EFSA Guidance document 2017. In their comparative analysis the authors compare 2 allergens and 2 non-allergenic proteins and follow their degradation via different conditions. This guidance document from EFSA J. 2017 states the refined /extended digestion assays are implemented for a time of 2 years to collect data that either support this part of weight of evidence approach or not – so this is a valuable information in that context.

The study is valid and highly relevant, state of the art, and provides data for the risk assessment discussion while it does not provide novel aspects.

The authors conclude that using different protocols for digestion assays and subsequent LC-MS detection does not provide additional information regarding allergenic risk assessment. Their conclusions is to stay with the established pepsin digestion assay which is sufficient for a weight of evidence approach for safety assessment of potential allergens.

What is missing is a discussion on peptides relevant for celiac disease where a) a library of peptides is available and b) the restriction of HLA types allows a thorough search for respective peptides. Has this been performed – or can this be discussed accordingly? Again this would be of relevance for the EFSA interim phase -see guidance document 2017. The discussion focuses on the enzyme kinetics and optimal conditions – true and relevant however, for physiological conditions in humans it may be a different scenario – this should be reflected in the discussion. Also the aspect of a potential protective role of phosphatidylcholine for digestion (Ref. Mandalari et al) should be discussed. Furthermore, as presented in the guidance document aspects on celiac disease and the relevant peptides should be included in the discussion.

Minor criticism:

What was the underlying rationale of selecting the non-allergenic protei

"I have read the journal's policy and the authors of this manuscript have the following competing interest: The authors are employees of Bayer CropScience, a leading manufacturer of crop seeds developed through conventional breeding or biotechnology. The funder had no role in study design, data collection and analysis, decision to publish, or preparation of the manuscript."
---

## [Author Response · Author response to Decision Letter 0]

30 Apr 2020

PONE-D-20-05235

Presence of small resistant peptides from new in vitro digestion assays detected by liquid chromatography-tandem mass spectrometry: an implication of allergenicity prediction of novel proteins?

PLOS ONE

Dear Dr. Maria Gasset

Academic Editor

PLOS ONE

Thank you for your recent letter with reviewer comments. Your recognition that our work is a valid and relevant contribution is much appreciated. All concerns are addressed after each reviewer’s comments in the attached original letter for easy of reading. The revised manuscripts with and without tracked changes have been submitted under file names of “Revised Manuscript with Track Changes” and “Revised Manuscript without Track Changes”, respectively. Files, containing names of “Cover Letter_COI”, original gel images and additional mass spectrometry data for “Supporting Information” have also been uploaded.

Please let me know if there are any questions.

Kind regards,

Rong Wang

Bayer Crop Science

PONE-D-20-05235

Presence of small resistant peptides from new in vitro digestion assays detected by liquid chromatography-tandem mass spectrometry: an implication of allergenicity prediction of novel proteins?

PLOS ONE

Dear Dr Ron Wang,

Thank you for submitting your manuscript to PLOS ONE and the time taking its review. The reviewer, who I agree with, has had problems in loading the comments and that is the reason I am copying them. We feel that it has merit but does not fully meet PLOS ONE’s publication criteria as it currently stands. Therefore, we invite you to submit a revised version of the manuscript that addresses the points raised during the review process.

"The study provided by the authors is according to state of the art and provides “real life data” comparing 2 allergens versus 2 non-allergenic proteins using the digestion assays with pepsin and chymotrypsin with and without bile salts following the EFSA Guidance document 2017. In their comparative analysis the authors compare 2 allergens and 2 non-allergenic proteins and follow their degradation via different conditions. This guidance document from EFSA J. 2017 states the refined /extended digestion assays are implemented for a time of 2 years to collect data that either support this part of weight of evidence approach or not – so this is a valuable information in that context.

The study is valid and highly relevant, state of the art, and provides data for the risk assessment discussion while it does not provide novel aspects.

The authors conclude that using different protocols for digestion assays and subsequent LC-MS detection does not provide additional information regarding allergenic risk assessment. Their conclusions is to stay with the established pepsin digestion assay which is sufficient for a weight of evidence approach for safety assessment of potential allergens.

What is missing is a discussion on peptides relevant for celiac disease where a) a library of peptides is available and b) the restriction of HLA types allows a thorough search for respective peptides. Has this been performed – or can this be discussed accordingly? Again this would be of relevance for the EFSA interim phase -see guidance document 2017. The discussion focuses on the enzyme kinetics and optimal conditions – true and relevant however, for physiological conditions in humans it may be a different scenario – this should be reflected in the discussion. Also the aspect of a potential protective role of phosphatidylcholine for digestion (Ref. Mandalari et al) should be discussed. Furthermore, as presented in the guidance document aspects on celiac disease and the relevant peptides should be included in the discussion.

Minor criticism:

What was the underlying rationale of selecting the non-allergenic proteins"

We would appreciate receiving your revised manuscript by May 15th. To enhance the reproducibility of your results, we recommend that if applicable you deposit your laboratory protocols in protocols.io, where a protocol can be assigned its own identifier (DOI) such that it can be cited independently in the future. For instructions see: http://journals.plos.org/plosone/s/submission-guidelines#loc-laboratory-protocols

• A rebuttal letter that responds to each point raised by the academic editor and reviewer(s). This letter should be uploaded as separate file and labeled 'Response to Reviewers'.

Response:

The rebuttal letter with a file name of “Response to Reviewers” has been uploaded.

• A marked-up copy of your manuscript that highlights changes made to the original version. This file should be uploaded as separate file and labeled 'Revised Manuscript with Track Changes'.

Response:

A revised manuscript with track changes is uploaded.

• An unmarked version of your revised paper without tracked changes. This file should be uploaded as separate file and labeled 'Manuscript'.

Response:

A revised manuscript without track changes is uploaded.

We look forward to receiving your revised manuscript.

Kind regards,

Maria Gasset, Ph.D.

Academic Editor

PLOS ONE

Additional Editor Comments (if provided):

The study provided by the authors is according to state of the art and provides “real life data” comparing 2 allergens versus 2 non-allergenic proteins using the digestion assays with pepsin and chymotrypsin with and without bile salts following the EFSA Guidance document 2017. In their comparative analysis the authors compare 2 allergens and 2 non-allergenic proteins and follow their degradation via different conditions. This guidance document from EFSA J. 2017 states the refined /extended digestion assays are implemented for a time of 2 years to collect data that either support this part of weight of evidence approach or not – so this is a valuable information in that context.

The study is valid and highly relevant, state of the art, and provides data for the risk assessment discussion while it does not provide novel aspects.

The authors conclude that using different protocols for digestion assays and subsequent LC-MS detection does not provide additional information regarding allergenic risk assessment. Their conclusions is to stay with the established pepsin digestion assay which is sufficient for a weight of evidence approach for safety assessment of potential allergens.

What is missing is a discussion on peptides relevant for celiac disease where a) a library of peptides is available and b) the restriction of HLA types allows a thorough search for respective peptides. Has this been performed – or can this be discussed accordingly? Again this would be of relevance for the EFSA interim phase -see guidance document 2017. The discussion focuses on the enzyme kinetics and optimal conditions – true and relevant however, for physiological conditions in humans it may be a different scenario – this should be reflected in the discussion. Also the aspect of a potential protective role of phosphatidylcholine for digestion (Ref. Mandalari et al) should be discussed. Furthermore, as presented in the guidance document aspects on celiac disease and the relevant peptides should be included in the discussion.

Response:

Thank you for your positive feedback. It is very encouraging that our work is recognized as valuable and relevant to the risk assessment discussion. Additional discussion text has been added to the revised manuscript to address the reviewer’s comments. 

Per reviewer’s first point, the insightful question aimed the exact essence of the relevance of 9 amino acid detection and is very much appreciated. Our additional discussion on peptide search and its relevance for non-IgE mediated celiac disease assessment is as follows:

Wheat gluten proteins, rich in proline and glutamine, and their allergenic epitopes are well characterized. For celiac disease, the 9 amino acid allergenic peptide has a distinct pattern of Q/E-X1-P-X2 motif, which can be presented by disease associated HLA-DQ (Human Leukocyte Antigen-DQ) molecules and can subsequently be recognized by T cells of celiac disease patients. A list of sequences containing this motif with their degenerated sequences was proposed by EFSA for sequence comparison (1). Such an evaluation is valuable and can inform on potential exposure to gluten or gluten-like peptides. A comparison of the amino acid sequences of all four proteins tested in the present study (i.e. �-Lg, �-La, Hb, and PFK) to the list of peptide sequences was conducted and revealed no relevant matches to known allergenic peptide sequences. Thus, evaluation of a test protein for the presence of allergenic peptide sequences can be done in silico, which has been routinely provided as part of the weight of evidence for the safety of newly expressed proteins in transgenic plants since early 2000 (CODEX, 2003). The combination of pepsin resistance analysis and in silico sequence homology analysis is sufficiently robust enough to address the question on protein stability against pepsin and sequence similarity to allergens. 

Per reviewer’s second point, physiological conditions in humans may indeed be a different scenario than the refined digestion conditions historically used. We pointed out in the Discussion (line 306 in original manuscript) that the physiological conditions of in vivo protein digestion are highly complex and constantly changing, such that no single in vitro model can fully represent, or simulate, the in vivo digestion processes. We also commented in the discussion (line 427 in original manuscript) that the overly simplified physiological digestion conditions through so-called refinement can neither truly reflect physiological conditions nor provide more reliable knowledge about the susceptibility of a NEP in the gastrointestinal tract. The in vitro digestibility assessment is useful to assess a protein’s physiochemical property at a define condition and provide attainable snapshot information.

Per reviewer’s third point, there is the aspect of a potential protective role of phosphatidylcholine (PC) for digestion (2). Mandalari et al observed that PC had no effect on pepsin digestion while it showed protection of b-Lg under duodenal digestion. Here are a few thoughts on her observations and why bile salts would be a relevant additive for a in vitro assay. First, Mandalari et al only tested one protein, b-Lg and concluded that PC had a protective role. Second, the critical micelle concentration (CMC) for PC is ~7 uM (2) vs 6 mM for sodium cholate or 16 mM for sodium deoxycholate which are two main components in bile salts. Low concentration of PC at uM are found in human gastric compartment (3). Therefore, the experimental concentration at 3 mM (2) well exceeded the CMC and physiological condition could form protection layers for proteins. Third, proteins in aqueous solution with low pH are expectedly unfolded and hydrolyzed during pepsin digestion while neutral pH and micelle formation could have enough protection on b-Lg during duodenal digestion. Finally, bile salts are major components of bile acid and suitable as a co-factor of intestinal digestion. 

We agree with reviewer that the relevant peptide should be evaluated rather than focus on peptide detection. This is clarified in the revised manuscript.

Minor criticism:

What was the underlying rationale of selecting the non-allergenic protein

Response: 

Hemoglobin (Hb) has been used to assess pepsin activity. We had the familiarity of Hb and studied Hb digested by pepsin at a broad pH range from 1.2 to 6 in a previous published paper (4). phosphofructokinase (PFK) was investigated by Astwood et al as pepsin labile protein (5) and recommended by EFSA as an assay control protein (1). Both Hb and PFK are non-allergens. Two allergens, a-La and b-Lg, a pepsin labile and a pepsin resistant, have been well studies. Both our experience and obtained literature knowledge can help us interpret data. Four proteins provided the representative spectra – digestible vs non-digestible, allergen vs non-allergen.

Response: 

Format of the manuscript has been updated to meet PLOS ONE requirement.

Response:

The original uncropped and unadjusted gel images are uploaded as Supporting Information.

Response:

A cover letter is updated to indicate original uncropped and unadjusted gel image files are in Supporting Information

"I have read the journal's policy and the authors of this manuscript have the following competing interest: The authors are employees of Bayer CropScience, a leading manufacturer of crop seeds developed through conventional breeding or biotechnology. The funder had no role in study design, data collection and analysis, decision to publish, or preparation of the manuscript."

Response:

The statement that "This does not alter our adherence to PLOS ONE policies on sharing data and materials” was confirmed and included in the cover letter.

Response:

I, the corresponding authors, declared, on behalf of all authors, all potential competing interests for the purposes of transparency 

Response:

Clarification was made to replace “data not shown”. Additional data with file name Supp_Table2.xlxs that contain number of peptides and peptide sequences was uploaded as Supporting Information 

References:

1. Naegeli H, Birch A, Casacuberta J, De Schrijver A, Gralak M, Guerche P, et al. Guidance on allergenicity assessment of genetically modified plants. EFSA Journal. 2017;15.

2. Mandalari G, Mackie AM, Rigby NM, Wickham MS, Mills EN. Physiological phosphatidylcholine protects bovine beta-lactoglobulin from simulated gastrointestinal proteolysis. Molecular nutrition & food research. 2009;53 Suppl 1:S131-9.

3. Minekus M, Alminger M, Alvito P, Ballance S, Bohn T, Bourlieu C, et al. A standardised static in vitro digestion method suitable for food - an international consensus. Food & Function. 2014;5(6):1113-24.

4. Wang R, Edrington TC, Storrs SB, Crowley KS, Ward JM, Lee TC, et al. Analyzing pepsin degradation assay conditions used for allergenicity assessments to ensure that pepsin susceptible and pepsin resistant dietary proteins are distinguishable. PloS one. 2017;12(2):e0171926.

5. Astwood JD, Leach JN, Fuchs RL. Stability of food allergens to digestion in vitro. Nature biotechnology. 1996;14(10):1269-73.

---

## [Decision Letter · Decision Letter 1]

13 May 2020

Presence of small resistant peptides from new in vitro digestion assays detected by liquid chromatography-tandem mass spectrometry: an implication of allergenicity prediction of novel proteins?

PONE-D-20-05235R1

Dear Dr. Rong Wang,

We are pleased to inform you that your manuscript has been judged scientifically suitable for publication and will be formally accepted for publication once it complies with all outstanding technical requirements.

With kind regards,

Maria Gasset, Ph.D.

Academic Editor

PLOS ONE

Additional Editor Comments (optional):

Reviewers' comments:

Reviewer's Responses to Questions

**Comments to the Author**

1. If the authors have adequately addressed your comments raised in a previous round of review and you feel that this manuscript is now acceptable for publication, you may indicate that here to bypass the “Comments to the Author” section, enter your conflict of interest statement in the “Confidential to Editor” section, and submit your "Accept" recommendation.

Reviewer #1: All comments have been addressed

2. Is the manuscript technically sound, and do the data support the conclusions?

Reviewer #1: Yes

3. Has the statistical analysis been performed appropriately and rigorously? 

Reviewer #1: Yes

4. Have the authors made all data underlying the findings in their manuscript fully available?

Reviewer #1: Yes

5. Is the manuscript presented in an intelligible fashion and written in standard English?

Reviewer #1: Yes

6. Review Comments to the Author

Reviewer #1: (No Response)

7. PLOS authors have the option to publish the peer review history of their article (what does this mean?). If published, this will include your full peer review and any attached files.

Reviewer #1: No

---

## [Editor Report · Acceptance letter]

26 May 2020

PONE-D-20-05235R1 

Presence of small resistant peptides from new *in vitro* digestion assays detected by liquid chromatography tandem mass spectrometry: an implication of allergenicity prediction of novel proteins? 

Dear Dr. Wang:

I am pleased to inform you that your manuscript has been deemed suitable for publication in PLOS ONE. Congratulations! Your manuscript is now with our production department. 

With kind regards,

on behalf of

Dr. Maria Gasset 

Academic Editor

PLOS ONE